# A LETM2-Regulated PI3K-Akt Signaling Axis Reveals a Prognostic and Therapeutic Target in Pancreatic Cancer

**DOI:** 10.3390/cancers14194722

**Published:** 2022-09-28

**Authors:** Shurui Zhou, Ziyi Zhong, Yanzong Lu, Yunlong Li, Hanming Yao, Yue Zhao, Tairan Guo, Kege Yang, Yaqing Li, Shaojie Chen, Kaihong Huang, Guoda Lian

**Affiliations:** 1Guangdong Provincial Key Laboratory of Malignant Tumor Epigenetics and Gene Regulation, Sun Yat-sen Memorial Hospital, Sun Yat-sen University, Guangzhou 510120, China; 2Department of Gastroenterology, Sun Yat-sen Memorial Hospital, Zhongshan School of Medicine, Sun Yat-sen University, Guangzhou 510120, China; 3Department of Ophthalmology, No.903 Hospital of PLA Joint Logistic Support Force, Hangzhou 310013, China

**Keywords:** LETM2, pan-cancer analysis, pancreatic ductal adenocarcinoma, PI3K-Akt pathway, tumor progression

## Abstract

**Simple Summary:**

LEMT2 was a newly discovered protein-encoding gene with little cancer research and an unclear mechanism. This study aimed to illustrate LETM2 as the crucial oncogene for tumor progression in pancreatic ductal adenocarcinoma (PDAC). We analyzed the expression level and prognostic value of LETM2 in multiple cancers using pan-cancer analysis and found that the LETM2 expression was the most significantly related to the dismal prognosis of PDAC. Immunohistochemical analyses showed that high LETM2 expression was correlated with poor outcomes of PDAC. In in vitro and in vivo experiments, LETM2 knockdown significantly inhibited tumor proliferation and metastasis, while LETM2 overexpression exerted the opposite effects. Then, we suggested that LETM2 may facilitate tumor progression by activating downstream PI3K-Akt signaling pathway in PDAC. In conclusion, the study enhanced our understanding of the LETM2-regulated PI3K-Akt signaling axis served as a prognostic and therapeutic target of pancreatic cancer.

**Abstract:**

Pancreatic ductal adenocarcinoma (PDAC) is one of the highest mortalities malignant tumors, which is characterized by difficult diagnosis, rapid progression and high recurrence rate. Nevertheless, PDAC responds poorly to conventional therapies, which highlights the urgency to identify novel prognostic and therapeutic targets. LEMT2 was a newly discovered protein-encoding gene with little cancer research and an unclear mechanism. Thus, this study aimed to illustrate LETM2 as the crucial oncogene for tumor progression in PDAC. In this study, we analyzed the expression level and prognostic value of LETM2 in multiple cancers using pan-cancer analysis. The analyses based on the TCGA-GTEx dataset indicated that the LETM2 expression was obviously elevated in several cancers, and it was the most significantly related to the dismal prognosis of PDAC. Subsequently, we demonstrated the functional role and mechanism of LETM2 by clinical sample evaluation, and in in vitro and in vivo experiments. Immunohistochemical analyses showed that high expression of LETM2 was correlated with poor outcomes of PDAC. Moreover, we demonstrated that LETM2 knockdown significantly inhibited tumor proliferation and metastasis, and promoted cell apoptosis, while LETM2 overexpression exerted the opposite effects. Finally, the impairment caused by LETM2-knockdown could be recovered via excitation of the PI3k-Akt pathway in vitro and in vivo animal models, which suggested that LETM2 could activate the downstream PI3K-Akt pathway to participate in PDAC progression. In conclusion, the study enhanced our understanding of LETM2 as an oncogene hallmark of PDAC. LETM2 may facilitate tumor progression by activating the PI3K-Akt signaling pathway, which provides potential targets for the diagnosis, treatment, and prognosis of pancreatic cancer.

## 1. Introduction

Pancreatic ductal adenocarcinoma (PDAC), emanating from the exocrine compartment, represents over 90% of all pancreatic malignancies [1]. Being among the most aggressive and lethal malignant solid tumors, PDAC is distinguished by rapid progression and high mortality, as evidenced by it having the lowest overall 5-year survival rate of 8% among all types of cancer [2,3,4]. PDAC bears an extremely dismal prognosis mainly because of its difficult early detection and suboptimal efficacy of traditional therapies [5]. Therefore, the exploration of the crucial molecular pathways and functional mechanisms driving the development as well as the progression of PDAC is critically required to discover new diagnostic biomarkers and identify novel therapeutic strategies in clinical practice.

LETM2 (leucine zipper-EF-hand-containing transmembrane protein 2, LETM2) belongs to the LETM1 protein family, which comprises LETM2, localized on chromosome 8p11.2, and its important paralog LETM1, encoded on chromosome 4p16.3 [6,7,8]. LETM1 and LETM2 are initially identified as nucleus-encoded, mitochondrial inner-membrane proteins. Mitochondrial activity is extremely important for maintaining all sorts of normal physiological functions, the malfunctioning of which may induce dysregulation of biological function or occurrence of various disorders such as malignant tumors [9,10].

It was discovered that elevated expression levels of LETM1 were related to poor prognosis in multiple malignancies [11,12]. Furthermore, LETM1 performs a vital role in driving development and metastasis by PI3K-Akt signaling pathway activation in ovarian cancer, prostate cancer, gastric cancer, and so on [13,14,15]. However, studies involving the biological roles and functional mechanisms of LETM2 in malignant tumors are scarce. LETM2 is located in close proximity to the Wolf–Hirschhorn syndrome candidate-1-like gene-1 (WHSC1L1), which is associated with Wolf–Hirschhorn syndrome, a rare genetic condition characterized by intellectual disability, delayed growth, psychomotor retardation and seizures [16]. It has been suggested that there is LETM2 amplification in lung cancer [17]. Cheng et al. detected the high-level amplification of LETM2 in esophageal squamous cell carcinoma by whole-genome sequencing, and downregulation of LETM2 expression inhibited tumor proliferation in vitro [18]. To date, there are no relevant studies concerning the LETM2 expression and its functional roles in PDAC.

Regarded as one of the most fundamental intracellular signal transduction pathways, the PI3K-Akt pathway is considered the master regulator of cell growth, apoptosis, migration, metabolism, and so on [19,20]. PI3K (phosphoinositide 3-kinase) is an upstream effector of the PI3K/Akt signaling pathway. By promoting the phosphorylation at Ser473 as well as Thr308, PI3K may induce the activation of critical downstream effector Akt. This pathway is triggered under physiological conditions when it is responsive to insulin, cytokines, and growth hormones to govern important metabolic activities. On the contrary, aberrant activation of PI3K-Akt and its downstream pathway contributes to the tumorigenesis and metastasis of malignant tumors [21,22]. PDAC is closely connected with Akt activation, which has been detected in 60% of PDAC cases [23,24]. Moreover, previous studies demonstrated a tough correlation between high PI3K-Akt pathway activity and poor prognosis of PDAC patients [25]. In recent years, inhibitors targeting the PI3K-Akt signaling pathway rapidly became under focus for clinical investigation in anti-cancer therapy, and encouraging evidence indicates the potential of PI3K-Akt drugs in both hematological malignancies as well as solid tumors [26,27,28].

In this study, we screened critical genes and pathways for pancreatic cancer progression using pan-cancer analysis and bioinformatics techniques. Then, we observed differential expression of LETM2 in different malignancies in the TCGA-GTEx pan-cancer dataset, which was most significantly associated with a dismal prognosis of PDAC. Subsequently, immunohistochemical analyses showed that the obviously elevated expression of LETM2 was correlated with abysmal outcomes of pancreatic cancer patients. Moreover, we demonstrated that the knockdown of LETM2 significantly promoted cancer cell apoptosis and inhibited tumor migration and proliferation in both in vivo and in vitro experiments. Finally, we suggested that LETM2 could activate the downstream PI3K-Akt pathway to participate in PDAC development. In conclusion, the study proposed that LETM2 may facilitate tumor development by promoting the PI3K-Akt signaling pathway in PDAC, which provides prospective targets for the diagnosis, treatment, and prognosis of PDAC. 

## 2. Materials and Methods

### 2.1. Evaluation of LETM2 Expression and Prognostic Significance in the Public Database

The TCGA pan-cancer dataset and the GTEx pan-cancer dataset containing normalized data were contacted and required from the UCSC Xena database [29] to conduct the pan-cancer analysis. Additionally, we downloaded RNA-sequencing expression profiles and relevant clinical information for 178 samples of pancreatic cancer from the TCGA dataset. The abbreviations and full names of cancers mentioned are listed in Appendix A. The limma package in R software was applied to make a comparison of the LETM2 expression between tumor and normal tissues of different malignancies, which may be validated by the GEPIA database [30]. We divided all patients into high and low groups on the basis of the median value of LETM2 expression in each cancer type to examine the impact of LETM2 expression on the overall survival (OS) as well as disease-free survival (DFS) in the survival analysis of PDAC and the pan-cancer analysis. Moreover, the GSVA package and ggstatsplot package in R software was used to present the correlations between LETM2 expression and signaling pathway as well as immune infiltration level. All the analysis methods and R packages were implemented by R version 4.0.3.

### 2.2. Tissue Specimens and Cell Culture

The paraffin-embedded PDAC specimens and matched non-tumor (paracancerous) tissue from 60 PDAC patients following the operation were collected from Sun Yat-sen Memorial Hospital between January 2016 and December 2021. Three kinds of PDAC cell lines (MIA PaCa-2, SW1990, and BxPc-3) and immortalized human pancreatic duct cells (hTERT-HPNE) were acquired from the American Type Culture Collection (ATCC, Manassas, VA, USA). The cells were cultivated in accordance with the manufacturers’ instructions. 

### 2.3. Cell Counting Kit-8 (CCK-8) Assays

CCK8 assays were performed to evaluate cell proliferation in vitro using the CCK8 Kit (Beyotime Biotechnology, Shanghai, China) on the basis of the manufacturer’s manual. In brief, cells were seeded in 96-well plates with 2 × 10^3^ cells per well. Then, 10 μL CCK8 solution was added to each well for incubation at 37 °C for 2 h. Finally, the absorbance was measured at 450 nm by a microplate reader.

### 2.4. Transwell Assays

With or without Matrigel (BD Biosciences, Franklin Lakes, NJ, USA), the PDAC cells were injected in the top transwell chamber (Corning, Corning, NY, USA) and incubated for 24 h or 48 h. Thereafter, cotton swabs were used to remove unmigrated and uninvaded PDAC cells. After being preserved in methanol for 10 to 15 min at room temperature, the PDAC cells should be stained with 0.5% crystal violet. Three triplicates of the tests were performed.

### 2.5. Quantitative Real-Time PCR (qRT–PCR)

Following the manufacturer’s directions, total RNA was extracted from tissues and cells using TRIzol Reagent (Invitrogen, Waltham, MA, USA). Using the PrimeScript RT reagent Kit and SYBR Premix Ex Taq (both from Takara, Japan), qRT–PCR was performed with the primer sequences shown in Appendix A. At least three times each of the trials were conducted.

### 2.6. Western Blotting (WB)

After extraction of total protein by using RIPA lysate (Solarbio, Beijing, China), the BCA Protein Assay Kit (Thermo Fisher Scientific, Waltham, MA, USA) was utilized to measure the protein concentrations. Equal amounts of proteins (20 μg) were separated on a 10% SDS/PAGE gel and electrotransferred (100 V, 2 h) to nitrocellulose membranes. The membranes were blocked with 5% BSA for 1 h and were then incubated with primary antibody for LETM2 (Proteintech, Wuhan, China) and GAPDH (Abcam, Cambridge, UK) overnight at 4 °C. The corresponding secondary antibodies were applied on the following day. The blots containing target bands were exposed by enhanced chemiluminescence (ECL) reagent (Solarbio, China) on the Exposure meter. The densitometry readings of each band were detected by ImageJ software 1.8.0 in the gray value analysis, and the relative expression was calculated as intensity ratio = target protein gray density/GAPDH gray density

### 2.7. Plasmid Construction

Small interfering RNA (siRNA) of LETM2 (siLETM2) and small hairpin RNA (shRNA) targeting LETM2 (shLETM2) were designed as shown in Appendix A and both synthesized by IGE Biotech (Guangzhou, China). To generate stable LETM2-knockdown cell lines, the PDAC cells were infected with lentiviral via transduction and then screened under 8 μg/mL puromycin pressure. Full-length LETM2 cDNA was designed and synthesized by IGE Biotech (Guangzhou, China), which was cloned into pcDNA3.1. 

### 2.8. Immunohistochemistry (IHC)

Tissue samples from PDAC patients as previously mentioned were fixed in formalin, embedded in paraffin, and cut into 4 μm sections. The paraffin-embedded sections were deparaffinized, hydrated, and blocked. Additionally, then the sections were incubated with primary antibodies overnight, followed by secondary antibodies for 2 h. Finally, DAB+ chromogen was used for color development and hematoxylin was applied to counterstain. IHC score for each case was calculated by the following formula: IHC score = positive percentage score × intensity score. Staining percentage score was graded as follows: 0; 0–25% of positively staining cells = 1; 25–50% of positively staining cells = 2; 50–75% of positively staining cells = 3; >75% of positively staining cells = 4. Staining intensity score was graded as follows: negative = 0; low positive = 1; positive = 2; high positive = 3. LETM2 expression was qualified as low (IHC score 0–6) and high (IHC score 8–12). 

### 2.9. Animal Experiments

The Animal Research Committee of the Sun Yat-sen University Cancer Center gave its approval to all animal trials. Female Balb/c nude mice (Balb/c-nu, weighing ~15–20 g), were purchased from the Guangdong Medical Laboratory Animal Center. A total of 20 mice were maintained under controlled environmental conditions in a 12 h light/dark cycle with unrestricted access to sterile feed and filtered water. We injected 5 × 10^6^ transfected BxPc-3 cells suspended in 0.2 ml of PBS into the flank of nude mice to construct the subcutaneous xenograft tumor model (five mice per group). Tumor size was surveyed once in 5 days (V = Length × Width^2^ × 0.5). The humane endpoints were defined as follows: (i) maximum tumor size > 1.5 cm^3^; (ii) weight loss >20%; (iii) tumor ulceration or necrosis. After four weeks, all of the nude mice in the study were sacrificed. The organs and tumors from the mice were dissected, photographed, weighed, and stained.

### 2.10. Statistical Analysis

The software involved in this study to perform statistical analysis contained SPSS 22.0 software, Stata 15.0 software, and R software 4.0.3. Student’s *t*-test was adopted for statistical analysis in the comparison of normally distributed continuous variables, while the Mann–Whitney U test was applied when our statistical data were not normally distributed. Additionally, categorical variables were processed by use of chi-square or Fisher exact test. Furthermore, we took advantage of Cox proportional hazards regression model for both univariate and multivariate analyses to calculate the hazard ratios (HR) along with associated 95% confidence intervals (CI). Variables with a *p*-value < 0.1 detected in univariate analyses were eligible for inclusion in the multivariate analyses. A value of *p* < 0.05 was supposed as an indication of a statistically significant difference.

## 3. Results

### 3.1. Pan-Cancer Analysis and Prognostic Value of LETM2

To assess the expression level and prognostic values of LETM2 in various cancers, pan-cancer analysis was implemented ground on the TCGA database and GTEx database. As indicated in Figure 1A,B, LETM2 was significantly elevated in 12 kinds of cancers compared to both paired adjacent tissues in TCGA and normal tissues in GTEx, including lung adenocarcinoma (LUAD, *p* = 5.7 × 10^−4^), colon adenocarcinoma (COAD, *p* = 7.0 × 10^−15^), colon adenocarcinoma/rectum adenocarcinoma (COADREAD, *p* = 1.4 × 10^−15^), breast invasive carcinoma (BRCA, *p* = 5.2 × 10^−13^), esophageal carcinoma (ESCA, *p* = 5.8 × 10^−3^), stomach and esophageal carcinoma (STES, *p* = 5.3 × 10^−6^), kidney renal papillary cell carcinoma (KIRP, p = 3.0 × 10^−4^), stomach adenocarcinoma(STAD, *p* = 8.4 × 10^−4^), head and neck squamous cell carcinoma (HNSC, *p* = 3.2 × 10^−10^), head and neck squamous cell carcinoma (LIHC, *p* = 3.8 × 10^−7^), rectum adenocarcinoma (READ, *p* = 0.03), pancreatic adenocarcinoma (PDAC, *p* = 0.02), pheochromocytoma/paraganglioma (PCPG, *p* = 3.4 × 10^−3^), cholangiocarcinoma (CHOL, *p* = 6.4 × 10^−6^), and cervical squamous cell carcinoma/endocervical adenocarcinoma (CESC, *p* = 0.05). Detail data of the statistical analysis are described in Appendix A. Then, we used the GEPIA database to further validate our results, which indeed revealed a significant increase in LETM2 expression in PDAC (Figure 1C). For exact values, see Appendix A. Moreover, pan-cancer survival analysis on the ground of the TCGA database was implemented to evaluate the predictive value and clinical significance of LETM2. The analysis in Figure 1D represented that elevated expression of LETM2 was most significantly associated with dismal prognosis of pancreatic cancer, either poor OS or DFS.

### 3.2. Prognostic Value and Clinical Significance of LETM2 for PDAC in TCGA Database

Therefore, we hereafter focused on the expression level and prognostic values of LETM2 in pancreatic cancer. The impact of LETM2 on OS and DFS of PDAC patients in the TCGA database was investigated. As presented in Figure 2A–C, the outcome showed that LETM2 overexpression negatively correlated with clinical outcome as OS (HR = 2.1, *p* = 0.00046) and DFS (HR = 1.7, *p* = 0.016) in PDAC patients. Subsequently, multivariate Cox regression analysis in TCGA database showed that LETM2 expression was an independent risk factor for OS of PDAC (Table 1). Additionally, we observed that LETM2 expression correlated with tumor inflammation signature (R = -0.15, *p* = 0.047), PI3K-AKT-mTOR pathway (R = −0.21, *p* = 0.006), and P53 pathway (R = 0.25, *p* = 0.001) (Figure 2D–F, Appendix A). On account of the relation between immune infiltration level and prognosis of malignancies, we further evaluated the correlation between LETM2 expression and immune infiltration level. As shown in Figure 2G,H and Appendix A, LETM2 expression correlated with immune infiltration level of B cell expression (R = −0.34, *p* = 3.95 × 10^−6^), and T cell CD8+ expression (R = 0.15, *p* = 0.045). The Sankey diagram in Figure 2I demonstrated the association between LETM2 expression and the histological grade of PDAC. The above findings suggested that LETM2 could bear the responsibility of a novel prognostic target for PDAC.

### 3.3. Elevated LETM2 Expression Correlates with Clinicopathological Features and Dismal Prognosis in PDAC Patients

To clarify the prognostic significance of LETM2 in PDAC patients, we first examined the LETM2 protein expression in six paired samples of PDAC tumor and non-tumor (paracarcinoma) tissues. The results showed that the LETM2 protein expression was remarkably increased in PDAC tumor tissues (Figure 3A,B). The densitometry readings/intensity ratio of each band and the original whole blot could be found in the Appendix A. The corresponding gene expression results of qRT–PCR also corroborated the WB findings, as indicated in Figure 3C. Moreover, IHC staining in specimens of 60 PDAC patients enrolled in our study indicated that LETM2 expression was significantly elevated in tumor tissue in comparison with the matched non-tumor tissue (n = 60, *p* < 0.001; Figure 3D,E). Then, we revealed significant positive correlations between IHC score and clinicopathological characteristics such as T stage, distant metastasis and pathologic stage, while there was no statistical significance between IHC score and lymph node metastasis (Figure 3F–H, Appendix A). Additionally, Kaplan–Meier survival analysis was put into practice to investigate the prognostic significance of LETM2 in PDAC. The results as showcased in Figure 3I,J found that elevated LETM2 expression was in connection with worse OS and DFS. In brief, these findings implied elevated LETM2 expression was significantly linked to poor prognosis in PDAC.

### 3.4. LETM2 Drives PDAC Tumorigenesis and Metastasis In Vitro

We evaluated the LETM2 expression in PDAC cell lines to investigate the functional roles of LETM2 in PDAC. In contrast to human pancreatic duct cells (hTERT-HPNE), the findings showed a considerable elevation of LETM2 mRNA as well as protein levels in PDAC cell lines (MIA PaCa-2, SW1990, and BxPc-3) (Figure 4A–C). In addition, consistent data was obtained in the analysis of the CCLE (Cancer Cell Line Encyclopedia) database (Appendix A). 

Therefore, we selected MIA PaCa-2 and SW1990 cells for subsequent experiments of LETM2 overexpression, and BxPc-3 and SW1990 cells for LETM2 knockdown experimental validation. We designed the overexpression lentivirus to construct overexpression cell lines named oeLETM2, while the names of the matching control cell lines were Ctrl. Moreover, BxPc-3 and SW1990 cell lines were transfected with siRNA of negative control (siNC) or targeting LETM2 (siLETM2-1, siLETM2-2 and siLETM2-3). The LETM2 expression of treated cells was confirmed with qRT–PCR and Western blotting. The findings suggested that the expression of LETM2 protein and mRNA was markedly increased in overexpression cell lines (Figure 4D–F), while both LETM2 protein and mRNA expression were significantly downregulated in knockdown cells (Figure 5A–C).

Firstly, we performed cell apoptosis analyses by flow cytometry. The results revealed that upregulation of LETM2 might lightly decrease cell apoptosis as demonstrated in Figure 4G, while the percentage of apoptotic cells was significantly higher in LETM2-knockdown PDAC cells (Figure 5D). Sequentially, we measured the growth of the above cells with LETM2 overexpression or knockdown by use of cell counting kit-8 (CCK-8) assays. Overexpression of LETM2 significantly promoted cell proliferation, and conversely (Figure 4I), LETM2 knockdown markedly led to a reduced trend in cell proliferation (Figure 5E). In addition, transwell assays were conducted to assess the migratory and invasive ability. As shown in Figure 4H, upregulation of LETM2 displayed a facilitation effect on not only migration but also invasion of PDAC cells. Additionally, the migratory and invasive ability of PDAC cells was prominently impeded after LETM2 knockdown (Figure 5F).

### 3.5. LETM2 Activates Downstream PI3K-Akt Signaling Axis in PDAC

To identify the target genes and key downstream pathways of LETM2, we performed a gene correlation analysis of LETM2 in the GEPIA database. Furthermore, the KEGG pathway and GO functional enrichment analyses were implemented on the top 100 co-expressed genes. According to the KEGG pathway enrichment analysis, several significantly enriched pathways (FDR < 0.1 and *p* < 0.05) were identified, including PI3K-Akt signaling pathway, axon guidance, small cell lung cancer, NF-kappa B signaling pathway, and so on (Figure 6A). Meanwhile, the molecular functions (MF) of GO functional enrichment analyses indicated signaling receptor binding, cadherin binding, extracellular matrix structural constituent, chemorepellent activity, etc. (Figure 6B). The biological processes (BP) of GO functional enrichment analyses indicated response to external stimulus, locomotion, cell adhesion, biological adhesion, etc. (Figure 6C). The cellular components (CC) of GO functional enrichment analyses indicated extracellular region, extracellular region part, vesicle, extracellular region space, cell junction, etc. (Figure 6D). 

Based on our experimental analysis and previous reports in the literature, we thought the PI3K-Akt signaling pathway served as the critical downstream pathway of LETM2 in PDAC. Then, Western blotting was conducted to validate the regulatory relationship between LETM2 and the PI3K-Akt pathway. As showcased in Figure 6E,F, the key proteins of the PI3K-Akt pathway including p-PI3K and p-Akt significantly downregulated in LETM2-knockdown cells, while there observed no discernible differences in PI3K and Akt. These results implied that LETM2 may activate downstream PI3K-Akt signaling pathway through enhancing the level of PI3K phosphorylation as well as Akt phosphorylation.

### 3.6. LETM2 Accelerates PDAC Malignant Progression by Activating the PI3K-Akt Signaling Pathway

In order to further demonstrate the biological functions and molecular mechanisms of LETM2-regulated PI3K-Akt signaling axis in PDAC, we used the PI3K-Akt pathway inhibitor and activator to conduct rescue experiments in vitro and in vivo. As indicated in Figure 7A, the flow cytometry results implied that inhibiting the PI3K-Akt pathway can also increase PDAC cell apoptosis, which was consistent with the trend of LETM2 downregulation. Additionally, meanwhile, the recovery of PI3K-Akt pathway activities via activator would partially rescue the cell apoptosis after LETM2 knockdown. Similarly, the PI3K-Akt pathway activator can abolish or decrease the effect of LETM2 knockdown including inhibiting PDAC cell proliferation, migration, and invasion during the rescue experiments (Figure 7B,C).

Consistent with the above observations, it was evident that the downregulation of LETM2 and inhibition of the PI3k-Akt pathway had a negative influence on tumor proliferation in the PDAC subcutaneous tumor model of nude mice. The temporal scheme of in vivo experiments was shown in Figure 8A. However, the impairment caused by LETM2-knockdown could be recovered via excitation of the PI3k-Akt pathway (Figure 8B–D). Furthermore, IHC staining of the nude mice tissues (Figure 8E) also verified that the Ki67 was downregulated in the LETM2 downregulation group and it was also reversed in the tumor tissues with the PI3k-Akt pathway activator stimulation. 

In conclusion, the findings of the present investigation confirmed that LETM2 accelerates PDAC malignant progression via activating the PI3K-Akt pathway.

## 4. Discussion

With a high mortality rate and rapid progression, PDAC remains one of the most treatment-refractory malignancies [4]. PDAC presents an extremely poor prognosis, and the incidence has been rising year by year. The 5-year OS rate of PDAC inferior to 3-10% is dismal in various studies [31,32,33]. However, the progress of diagnosis and treatment in PDAC is slow on account of its complexity and difficulty [34,35]. Meanwhile, PDAC treatment is still a significant challenge due to the poor efficacy of the current therapies. Surgical resection continues to be the sole means for a cure in patients with PDAC to date, but the majority are diagnosed too late for surgery due to the lack of specific diagnostic indicators. Many patients still experience relapse even after surgery and chemotherapy [36]. Therefore, the mechanism of pancreatic cancer should be further explored and it is of great significance to discover effective diagnostic and therapeutic targets.

In the current research, we firstly identified LETM2 as an oncogenic hallmark of PDAC by pan-cancer analysis of the TCGA + GTEx database and the results of GEPIA analysis also supported this finding. Pan-cancer survival analysis ground on the TCGA database was implemented to evaluate the predictive value and clinical significance of LETM2, which represented that elevated expression of LETM2 was most significantly associated with dismal prognosis of pancreatic cancer, either poor OS or DFS. Moreover, we hereafter focused on the prognostic values of LETM2 in pancreatic cancer. PDAC patients with elevated LETM2 expression had substantially shorter OS times than those with low LETM2 expression. Additionally, the ROC curve of LETM2 indicated a high predictive power. Furthermore, IHC staining in specimens of 60 PDAC patients enrolled in our study verified that LETM2 expression was significantly elevated in tumor tissue in comparison with the matched non-tumor tissue. Additionally, there revealed significant positive correlations between IHC score and clinicopathological characteristics such as stage and distant metastasis. Additionally, elevated LETM2 expression was significantly linked to poor prognosis in PDAC. The above findings suggested that LETM2 could bear the responsibility of a novel prognostic target for PDAC.

LETM2 was first found within a region on chromosome 8p11.2. LETM2 is a kind of protein-coding gene exhibiting high sequence similarity to LETM1. The 8p11.2 region comprises a cluster of genes containing LETM2, FGFR1, and WHSC1L1 [16]. Rearrangements of chromosome 8p are continually reported in various types of cancers, which were revealed to be tightly connected with malignant tumor development and progression [37,38]. There have been many reports about the close relationship between FGFR1 and cancer development [39,40,41]. In addition, several studies focused on the oncogenic role of WHSC1L1 in breast cancer, pancreatic adenocarcinoma, and lung tumor in recent years [42,43]. However, although there is differential expression of LETM2 in multiple cancers, research on the function and mechanism of LETM2 in tumor development and progression is rare. The prognostic value and clinical significance of LETM2 need to be further explored. 

On the basis of this perspective, we next demonstrated the functional role and mechanism of LETM2 by in vitro and in vivo trials. In comparison to normal pancreatic duct cells, LETM2 mRNA and protein expression were observed to be markedly elevated in PDAC cells. LETM2 knockdown could not only suppress PDAC cell proliferation, migration, and invasion, but also promote PDAC cell apoptosis in vitro, while LETM2 overexpression may speed up tumor development and progression. Additionally, the tumorigenicity of PDAC cells was decreased by the knockdown of the LETM2 in vivo xenograft tumor model of nude mice. The above-mentioned results presented the important role of LETM2 indicating the prognosis and promoting tumor progression.

To identify the target genes and key downstream pathways of LETM2, we performed gene correlation analysis of LETM2 in the GEPIA database. Furthermore, the KEGG pathway and GO functional analyses were implemented on the top 100 co-expressed genes. According to the KEGG analysis, we considered the PI3K-Akt pathway as the critical downstream pathway of LETM2 in PDAC. Then, Western blotting was conducted to validate the regulatory relationship between LETM2 and the PI3K-Akt pathway. The key proteins of the PI3K-Akt pathway including p-PI3K and p-Akt significantly downregulated in LETM2-knockdown cells, while no discernible differences in PI3K and Akt were observed. These results implied that LETM2 may activate the downstream PI3K-Akt signaling pathway through enhancing the level of PI3K phosphorylation as well as Akt phosphorylation.

Dysregulation of the PI3K-Akt signaling pathway appears as one of the most frequent oncogenic events related to tumor progression in all kinds of malignancies [44,45]. The main downstream effectors of PI3K are AKT and mTOR (mammalian targets of rapamycin), and the pathway bears a fundamental oncogenic role in driving tumor evolution and progression in pancreatic cancer. Inhibitors aimed at the pathway could provide novel and important strategies for the treatment of pancreatic cancer [28,46]. 

Therefore, we used PI3K inhibitor GDC-0941 [47] and AKT activator SC-79 [48] to conduct rescue experiments in vitro as well as in vivo, in which we noticed that the PI3K-Akt pathway activator can abolish or decrease the effect of inhibiting tumor progression after LETM2 knockdown. Consistent with the above observations, it was evident that the downregulation of LETM2 and inhibition of the PI3k-Akt pathway had a negative influence on tumorigenesis in the PDAC subcutaneous xenograft tumor model of nude mice. The impairment caused by LETM2-knockdown could be recovered via activation of the PI3k-Akt pathway. 

However, we noticed that knockdown LETM2 can suppress PDAC malignant progression more efficiently than Akt inhibitors. Additionally, after the recovery of the PI3K-Akt pathway by the activator, both the proliferation and metastasis ability of PDAC cells were incapable to return to the original level completely, which dropped a hint that LETM2 may adjust tumor progression by means of diverse signaling pathways besides the PI3K-Akt pathway in PDAC. Therefore, LETM2 may act as not only a prognostic but also a therapeutic target for early diagnosis and novel treatment of PDAC. We will be pursuing the feasibility of inhibitors targeting LETM2 such as delivering siLETM2s by smart nanoparticles in subsequent studies.

## 5. Conclusions

To summarize, we observed differential expression of LETM2 in various malignancies in the pan-cancer analysis, which was most prominently correlated with the dismal prognosis of PDAC. Results of in vivo as well as in vitro trials confirmed that LETM2 accelerates PDAC malignant progression via activating the downstream PI3K-Akt pathway. This is the first report on the functional role and molecular mechanism of the newly protein-coding gene, LETM2, in pancreatic cancer. We identified the LETM2-regulated PI3K-Akt signaling axis as the crucial element of PDAC development and progression, which provides effective targets for the diagnosis, treatment, and prognosis of PDAC.

## Figures and Tables

**Figure 1 cancers-14-04722-f001:**
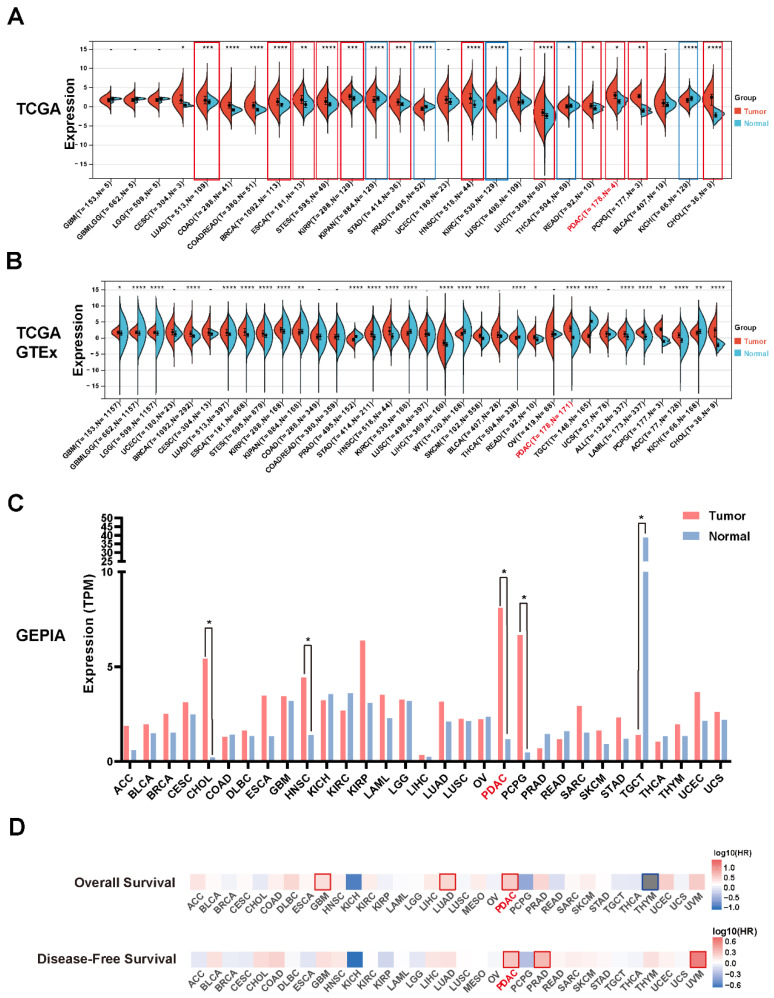
Expression of LETM2 and its prognostic value in pan-cancer analysis and PDAC. (**A**) Comparison of LETM2 expression in tumor tissue and adjacent normal tissue in the TCGA database. Cancers indicated in red or blue denotes have significantly different expressions of LETM2 in comparison with tumor tissues and normal tissues. (**B**) Comparison of LETM2 expression in tumor tissue and normal tissue in samples combined TCGA database and GTEx database. The abbreviations and full name of cancer mentioned in (**A**) and (**B**) are listed in Appendix A. (**C**) Comparison of LETM2 expression in tumor tissue and normal tissue in the GEPIA database. Cancers indicated in red or blue denotes have significantly different expressions of LETM2 in comparison with tumor tissues and normal tissues. (**D**) Pan-cancer survival analysis of LETM2 based on the TCGA database. Statistical analysis was compared to the control or normal groups: * *p* < 0.05; ** *p* < 0.01; *** *p* < 0.001; **** *p* < 0.0001.

**Figure 2 cancers-14-04722-f002:**
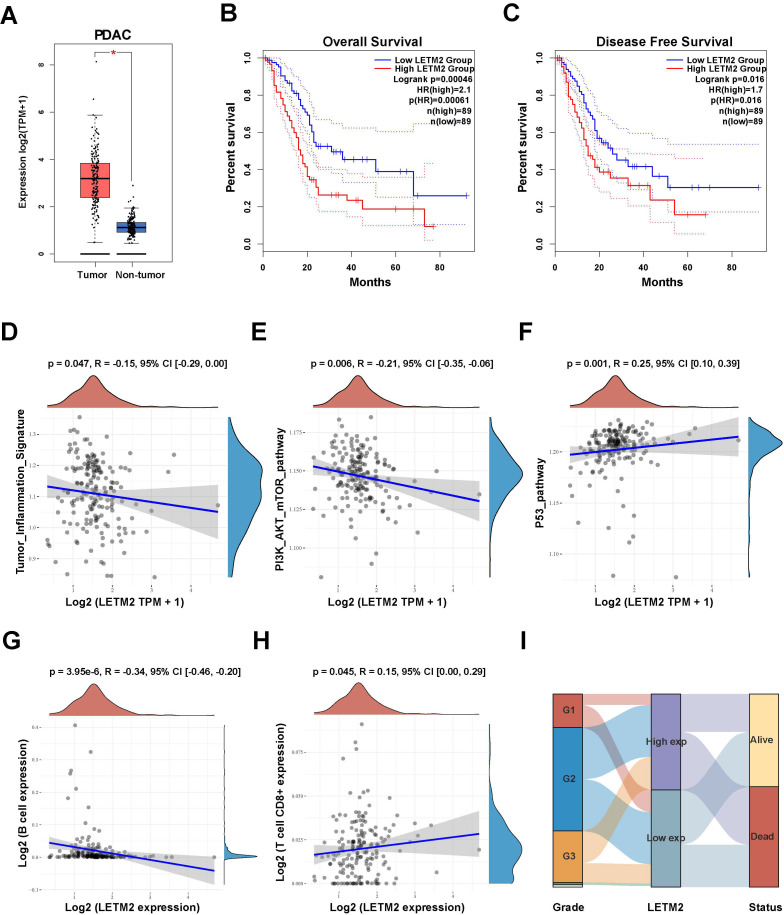
Validation of LETM2 prognostic value and clinical significance for pancreatic cancer in TCGA database. (**A**) TCGA + GTEx database analysis revealed that LETM2 expression was significantly elevated in PDAC tumor tissue in comparison with normal tissue. Red represents tumor tissue and blue represents normal tissue. (**B**,**C**) High levels of LETM2 expression correlated with significantly poor (**B**) OS and (**C**) DFS of PDAC patients based on survival analysis of the TCGA database. The dotted lines represented 95% confidence intervals. (**D**) LETM2 expression correlated with tumor inflammation signature. (**E**) LETM2 expression correlated with PI3K-AKT-mTOR pathway. (**F**) LETM2 expression correlated with the P53 pathway. (**G**) LETM2 expression correlated with immune infiltration level of B cell expression. (**H**) LETM2 expression correlated with immune infiltration level of T cell CD8+ expression. (**I**) Sankey diagram demonstrated the association between LETM2 expression and histological grade and living status. * *p* < 0.05.

**Figure 3 cancers-14-04722-f003:**
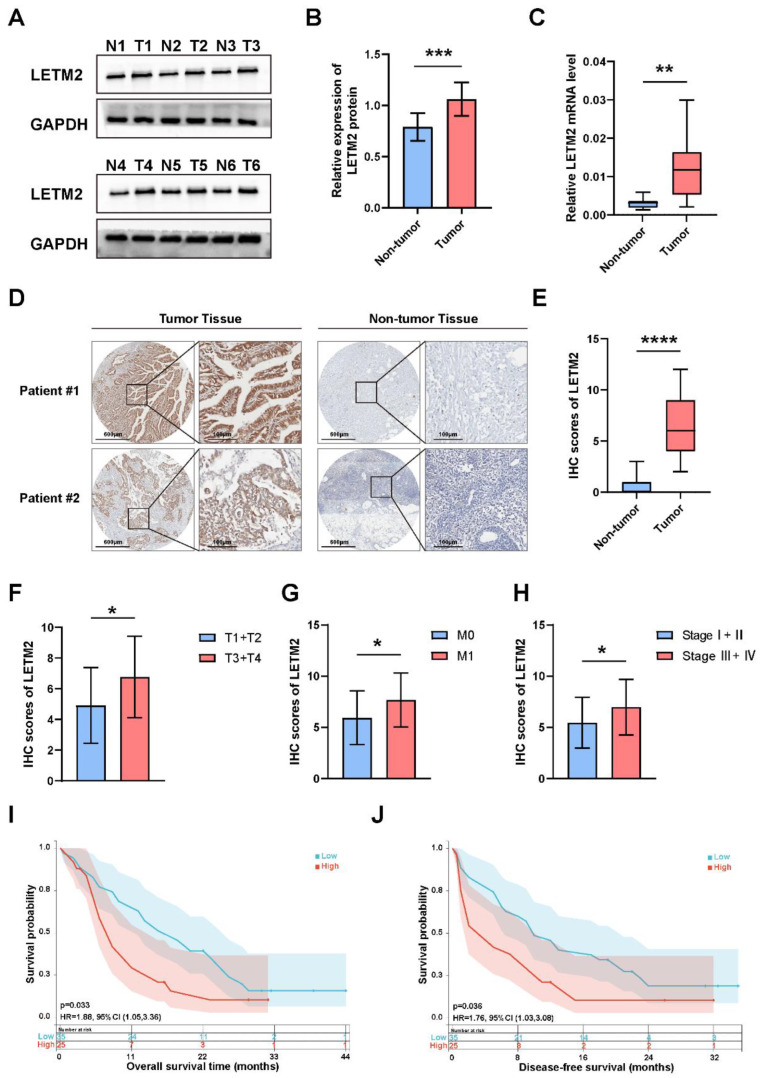
Elevated LETM2 expression correlates with clinicopathological features and poor prognosis in PDAC patients. (**A**) The protein expression level of LETM2 was examined by Western blotting in paired samples of PDAC tumor and non-tumor tissues. The densitometry readings/intensity ratio of each band and the original whole blot could be found in Appendix A. (**B**) Quantitative analysis of Western blotting and (**C**) qRT-PCR analysis revealed LETM2 protein expression was remarkably increased in PDAC tumor tissues. The two-tailed *t*-test in paired samples. (**D**) Representative images of IHC staining for LETM2 protein of PDAC tumor tissues and their adjacent non-tumor tissues. Scale bars, left: 600 μm, right: 100 μm. (**E**) Quantification of IHC staining score. (**F**–**H**) Relationship between IHC score of LETM2 and clinicopathological features of PDAC patients, such as (**F**) T stage, (**G**) distant metastasis and (**H**) pathological stages. Overall survival (**I**) and disease-free survival (**J**) analyses in PDAC patients in the high LETM2 expression group and low LETM2 expression group were conducted by Kaplan–Meier method with the two-tailed log-rank test. The data are represented as mean ± SEM, and Student’s *t*-test was performed. ns, not significant; * *p* < 0.05; ** *p* < 0.01; *** *p* < 0.001; **** *p* < 0.0001.

**Figure 4 cancers-14-04722-f004:**
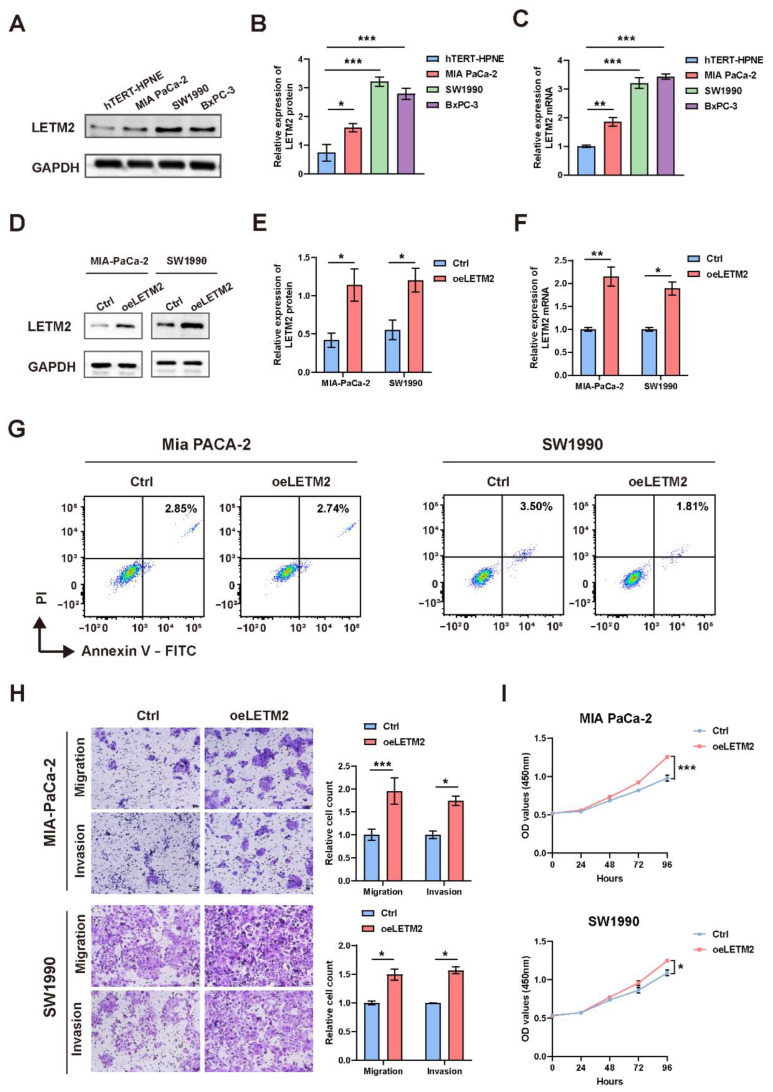
LETM2 promotes PDAC cell proliferation, migration, and invasion in vitro. (**A**) Western blotting analysis of LETM2 protein expression in three types PDAC cell lines (MIA PaCa-2, SW1990, and BxPc-3) compared with human pancreatic duct cells (hTERT-HPNE). The densitometry readings/intensity ratio of each band and the original whole blot could be found in Appendix A. (**B**) Quantitative analysis of Western blotting and (**C**) qRT-PCR analysis revealed LETM2 protein and mRNA expression were significantly higher in PDAC cell lines. (**D**) Western blotting analysis for verifying the efficiency of LETM2 overexpression. The densitometry readings/intensity ratio of each band and the original whole blot could be found in Appendix A. (**E**) Quantitative analysis of Western blotting and (**F**) qRT-PCR analysis revealed that LETM2 protein and mRNA expression were significantly elevated in overexpression cells. oeLETM2 indicates MIA PaCa-2 cells and SW1990 cells transfected with an overexpressing plasmid. The relative quantification was calculated using the 2^−ΔΔCt^ method and normalized based on GAPDH. (**G**) Cell apoptosis of LETM2 overexpression cell lines was measured using flow cytometry. (**H**) Cell migratory and invasive ability of LETM2 overexpression cell lines were measured using transwell assay. Representative photographs (**left**) and quantification (**right**) were shown. Scale bars, 50 μm. (**I**) Cell growth of LETM2 overexpression cell lines was measured using CCK-8 assays. The data are represented as mean ± SEM (n = 3 biologically independent experiments), and Student’s *t*-test was performed. * *p* < 0.05; ** *p* < 0.01; *** *p* < 0.001.

**Figure 5 cancers-14-04722-f005:**
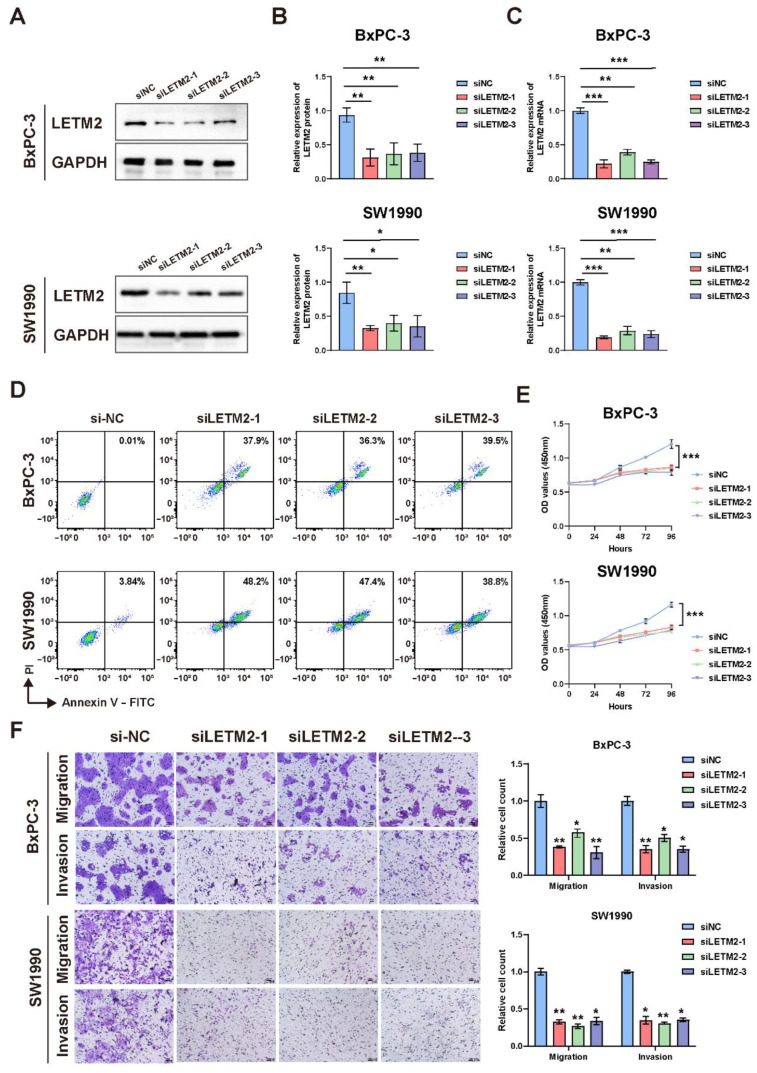
Downregulation of LETM2 inhibits PDAC cell proliferation and metastasis and promoted cell apoptosis in vitro. (**A**) Western blotting analysis for verifying the efficiency of LETM2 knockdown. The densitometry readings/intensity ratio of each band and the original whole blot could be found in Appendix A. (**B**) Quantitative analysis of Western blotting and (**C**) qRT-PCR analysis revealed LETM2 protein and mRNA expression were significantly downregulated in knockdown cells. siLETM2 indicates BxPc-3 cells and SW1990 cells transfected with siRNA targeting LETM2. The relative quantification was calculated using the 2^−ΔΔCt^ method and normalized based on GAPDH. (**D**) Cell apoptosis of LETM2 knockdown cell lines was measured using flow cytometry. (**E**) Cell growth of LETM2 knockdown cell lines was measured using CCK-8 assays. (**F**) Cell migratory and invasive ability of LETM2 knockdown cell lines were measured using transwell assay. Representative photographs (**left**) and quantification (**right**) were shown. Scale bars, 50 μm. The data are represented as mean ± SEM (n = 3 biologically independent experiments), and Student’s *t*-test was performed. * *p* < 0.05; ** *p* < 0.01; *** *p* < 0.001.

**Figure 6 cancers-14-04722-f006:**
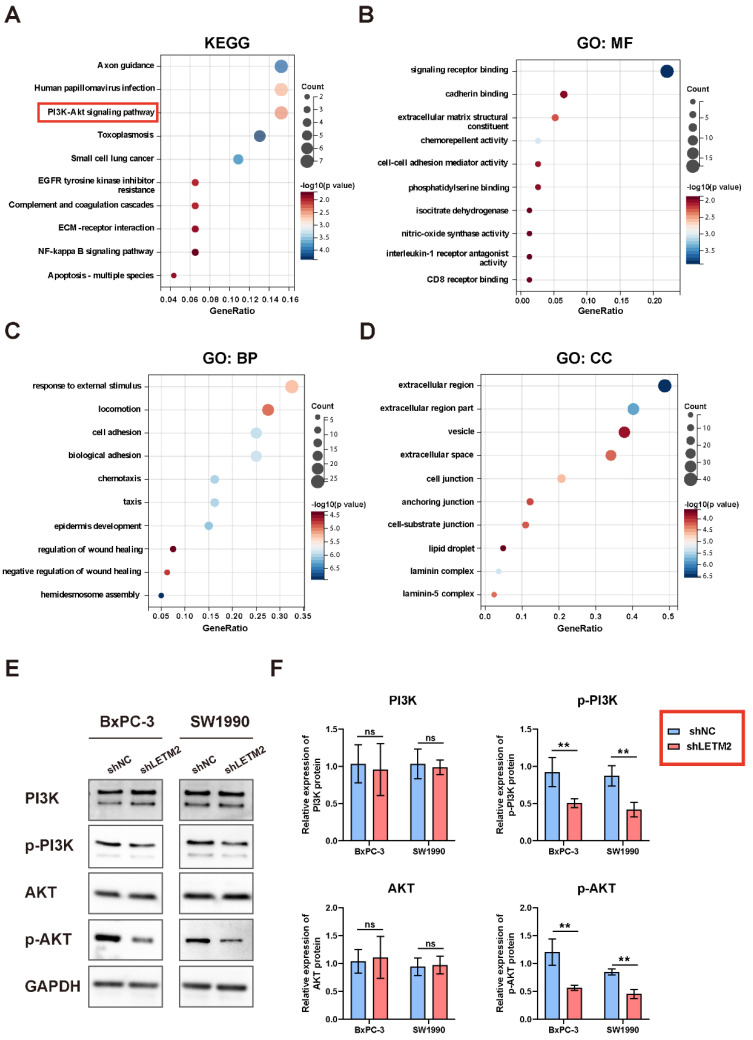
LETM2 may activate the downstream PI3K-Akt signaling pathway by increasing the phosphorylation of PI3K and Akt. (**A**) KEGG pathway enrichment analyses were performed on LETM2 and co-expressed genes in PDAC. (**B**) The molecular functions (MF) of GO functional enrichment analyses were performed on LETM2 and co-expressed genes in PDAC. (**C**) The biological processes (BP) of GO functional enrichment analyses were performed on LETM2 and co-expressed genes in PDAC. (**D**) The cellular components (CC) of GO functional enrichment analyses were performed on LETM2 and co-expressed genes in PDAC. (**E**) Western blotting analysis for evaluating the key protein level of PI3K-Akt signaling pathway after LETM2 knockdown. The densitometry readings/intensity ratio of each band and the original whole blot could be found in Appendix A. (**F**) Quantitation of Western blotting implied that p-PI3K and p-Akt were significantly inhibited after LETM2 knockdown. Student’s *t*-test was performed. ns, not significant; ** *p* < 0.01.

**Figure 7 cancers-14-04722-f007:**
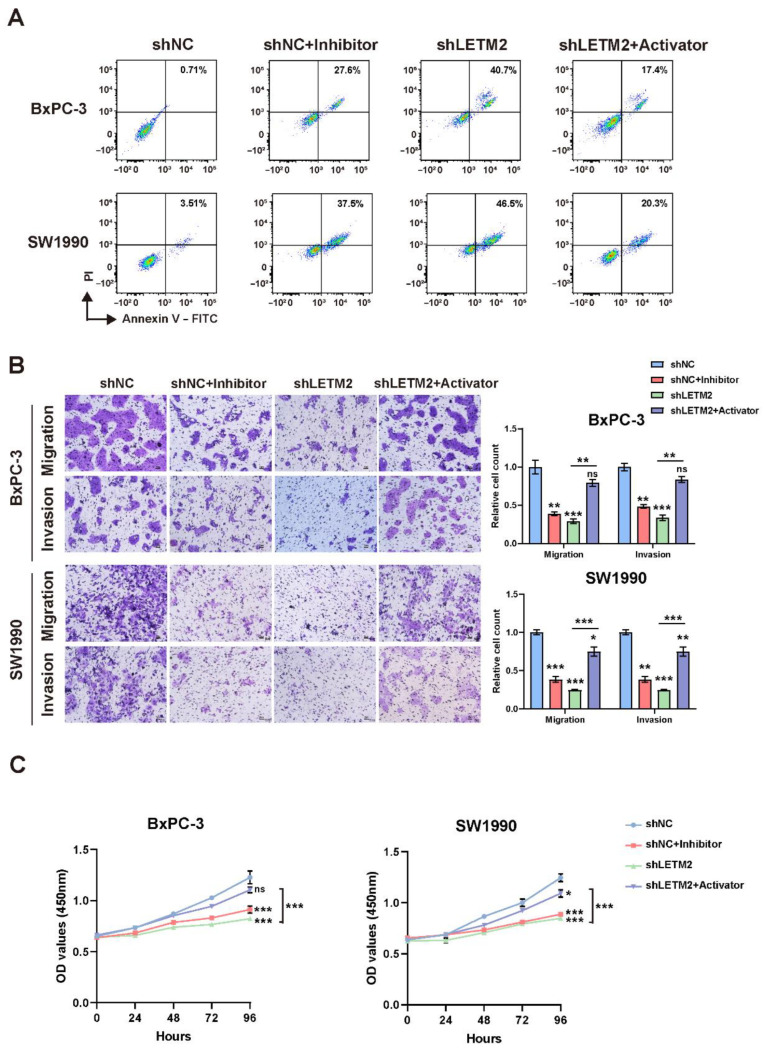
LETM2 accelerates PDAC malignant progression by activating the PI3K-Akt pathway in vitro. (**A**) Cell apoptosis was measured using flow cytometry. (**B**) Cell migratory and invasive ability were measured using transwell assay. Representative photographs (**left**) and quantification (**right**) were shown. Scale bars, 50 μm. (**C**) Cell growth was measured using CCK-8 assays. The data are represented as mean ± SEM (n = 3 biologically independent experiments), and Student’s *t*-test was performed. Ns, not significant; * *p* < 0.05; ** *p* < 0.01; *** *p* < 0.001.

**Figure 8 cancers-14-04722-f008:**
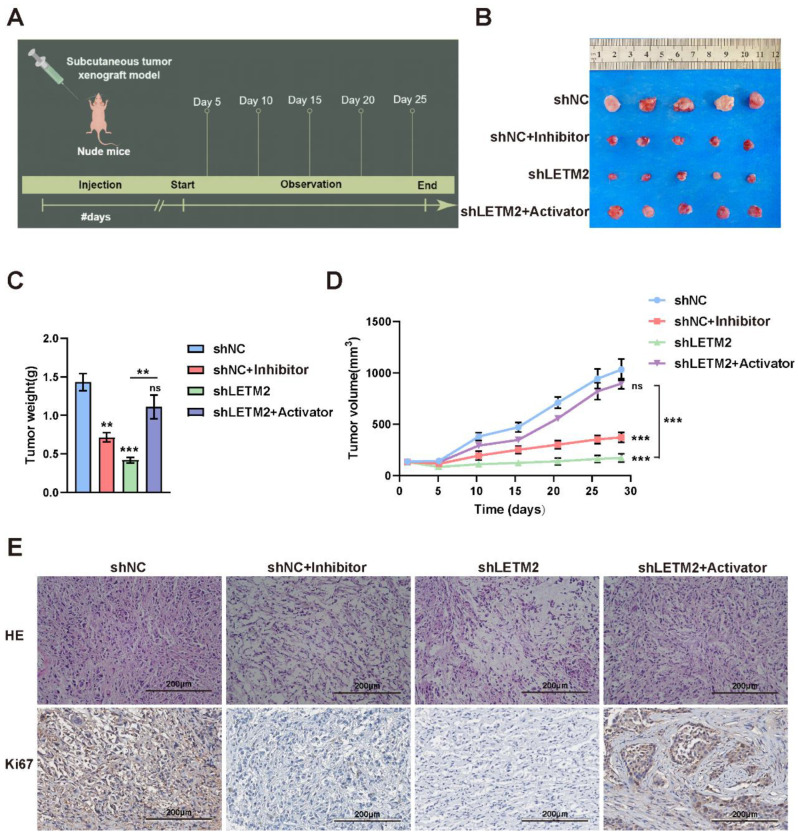
LETM2 accelerates PDAC malignant progression by activating the PI3K-Akt pathway in vivo. (**A**) Temporal scheme of in vivo experiments. (**B**) Gross images of PDAC subcutaneous tumor. (**C**) The tumors were extracted and weighed after 4 weeks. (**D**) The tumor volume was monitored every five days, and tumor growth curves were generated. (**E**) H&E and IHC staining with an antibody specific for Ki67 in sections of tumors. The data shown were representative of three experiments. Scale bars, 200 μm. Student’s *t*-test was performed. ns, not significant; ** *p* < 0.01; *** *p* < 0.001.

**Table 1 cancers-14-04722-t001:** Univariate and multivariate Cox regression analyses of overall survival of PDAC in TCGA database.

Characteristics	n	Univariate Analysis		Multivariate Analysis	
		Hazard Ratio (95% CI)	*p* Value	Hazard Ratio (95% CI)	*p* Value
Age	178				
≤65	93	Reference			
>65	85	1.290 (0.854–1.948)	0.227		
Gender	178				
Female	80	Reference			
Male	98	0.809 (0.537–1.219)	0.311		
T stage	176				
T1 + T2	31	Reference			
T3 + T4	145	2.023 (1.072–3.816)	0.030 *	1.322 (0.658–2.654)	0.433
N stage	173				
N0	50	Reference			
N1	123	2.154 (1.282–3.618)	0.004 *	2.071 (1.179–3.637)	0.011 *
M stage	84				
M0	79	Reference			
M1	5	0.756 (0.181–3.157)	0.701		
Pathologic stage	175				
Stage I + II	167	Reference			
Stage III + IV	8	0.673 (0.212–2.135)	0.501		
Histologic grade	176				
G1 + G2	126	Reference			
G3 + G4	50	1.538 (0.996–2.376)	0.052	1.196 (0.764–1.871)	0.434
LETM2 expression	178	1.524 (0.989–2.350)	0.056	1.658 (1.052–2.613)	0.030 *

CI, confidence interval; variables with a *p*-value < 0.1 in univariate analyses were included in the multivariate analyses. * *p* < 0.05 indicates significance.

## Data Availability

The data presented in this study are available on request from the corresponding author.

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
