# Peer review of "A LETM2-Regulated PI3K-Akt Signaling Axis Reveals a Prognostic and Therapeutic Target in Pancreatic Cancer"

_cancers, 2022, doi:10.3390/cancers14194722_

Round 1
Reviewer 1 Report
The authors describe LETM2 as a potential oncogene relevant in PDAC that correlates in its expression and function with OS and DFS of human PDAC patients. This function is demonstrated by gain-of-function and loss-of-function experiments. LETM2 acts probably via the PI3-K-Akt signaling axis. This could be of potential interest for translational oncology. However the manuscript suffers from many insufficiencies.
Major points:
1. The Abstract is too general as compared to the Simple Summary. There is nearly no additional distinct information given.
2. The rationale for the selection of the cell lines is not described: What is their K-Ras status and why did you choose these three cell lines?
3. Most important critisism relates to the inconsistent and incomplete presentation of proliferation, metastasis and apoptosis assays for all three cell lines in vitro. While the xenograft in vivo assay is done with BxPC3 only, this cell line was not used in vitro for proliferation and invasion/migration assays in the OE-setup. In the KD -setup MiaPaCa-2 cells were not included. A complete set of in vitro and in vivo data for all three cell lines needs to be presented to follow the conclusions.
It is stated that the OS correlates with LETM2expression. Please present Kaplan-Maier Plots of your experiment with the OE and KD-cell lines.
The same inconsistency is given with the PI3K-Akt axis analysis.
4. Figure 8: The tumor volume data for day 28 are missing. Why?
5. Please define the humane endpoints of your tumor experiment under Point 2.7. Which diet was used to feed the mice and please specify the conditions of day and night-rhythm.
Figure 3A-B: The expression level of LETM2 in 3 tumor and 3 paired normal samples are quantified under 3B. It seems that the GAPDH-signal intensity is not in the linear range of grey and therefore the quantification is questionable. Furthermore a two tailed t-test of N=3 samples seems not meaningful. Please increase the number of samples or just describe the result.
Minor points:
1. Two different abbrivations are used for pancreatic ductal adenocarcinoma, i. e. PDAC and PAAD. Please change.
2. What is the difference between MiaPaCa-2 and MiaPaCa-3 (SFig. 3) ?
3. Abstract: Is that really correct as stated that PDAC has the highest incidence of malignant tumors? Please check again.
4. Abstract: Please explain the Wolf-Hirschhorn syndrome.
5. The nature of the control tissue is unclear: is it comparable normal from healthy or inflamed pancreas donors or matched tissue of the same tumor?
Reviewer 2 Report
Dr. Zhou et al provide new insight on the relationship between LETM2 and pancreatic cancer. The authors consider LETM2 in multiple cancers using pan-cancer analysis, conduct immunohistochemical analyses on the impact of LETM2 on outcomes in PDAC, conduct in vitro and in vivo experiments to evaluate the impact of LETM2 on tumor proliferation, etc., and consider the signaling pathway in PDAC. I believe this article can potentially be a value addition to the existing literature. My concerns are as follows.
Major points:
1. This study essentially contains several studies in one. Each requires detailed explanations, which are not provided in the Materials and Methods. There are only 5-6 lines for each portion. In addition, baseline data is not provided for much of the analyses, making it impossible for the reader to reproduce results.
2. The study is difficult to follow because the Methods, Results, and Discussion are mingled, with some methods explained under Results, some interpretations which should be included in the Discussion included in Results, etc. Please ensure that explanations are in the correct section.
3. The pan-cancer analysis may be of value, but is not central to this study. In particular, the authors spend too much time and space discussing cancers other than PDAC, which is the focus of this study. This section can be shortened. Figure 1C is just copied and pasted from GEPIA and does not need to be included.
4. Figure 2: Columns should be labeled in Figure 2A to facilitate viewing. What is the value in analyzing HNSC? The authors should stay focused on PDAC. Thus, Figure 2C and 2D are not necessary. Figure 2B and 2D: are the dotted lines confidence intervals? This should be explained. Is number at risk available for Figure 2B? Figure E: Can you revise the x axis to months, to conform with Figure 2B? Using the same labels and ticks for the y axis can also be helpful. Figure 2F: The text says that AUC of 0.704 shows “high predictive power”, but I do not believe this is a high figure. Also, the AUCs for 3 years and 5 years is less than 0.5?
5. Figures 2B and 2E: Can you provide baseline characteristics of the pancreatic cancer patients in the GEFIA database and the ICGC database in supplementary tables? Otherwise the reader cannot evaluate confounding factors; the data comes out of nowhere and the results of the research cannot be replicated. If background data is available, please also conduct univariate and multivariate Cox regression analyses to see if LETM2 expression is an independent predictor of OS and DFS in PDAC patients.
6. Figure 3: How were IHC scores calculated? Also, the value of Figures 3F-3H is unclear because the reader cannot tell whether higher IHC scores are associated with more advanced disease or if the baseline characteristics were different. Is Table S5 a table of baseline characteristics between the high and low groups? If so, more data should be provided because various confounders such as surgery, treatment regimen, recurrence, etc. are not considered. P-values should also be provided, along with a Cox regression analysis. Without these, the authors cannot conclude that “elevated LETM2 expression was significantly linked to poor prognosis in PDAC” (page 7).
Similarly, Figure 3I, 3J are also of questionable value because baseline characteristics are not available.
Minor points:
1. There are multiple references to “PADC”. Did the authors mean PDAC?
2. Figure 1: Statistical significance is shown by one, two, three, or four stars in Figure 1A and 1B. However, the legend only refers to one star or three stars: “Statistical analysis was compared to the control or normal groups: * p < 0.05; *** p < 0.001.”
3. Some portions in the Results section, such as the first half of 3.3, should be moved to Materials and Methods.
4. There are two tables with the title “Table S5.”
5. I did not understand the second Table S5: “The SH3TC2 expression and the clinical characteristic of CRC patients.” Do the authors mean PDAC patients? The IRB statement also refers to “all enrolled CRC patients.” Why suddenly SH3TC2?
Round 2
Reviewer 2 Report
The authors have adequately revised their manuscript.